# Carbon Fibers from High-Density Polyethylene Using a Hybrid Cross-Linking Technique

**DOI:** 10.3390/polym13132157

**Published:** 2021-06-30

**Authors:** Seong-Hyun Kang, Kwan-Woo Kim, Byung-Joo Kim

**Affiliations:** 1Research and Development Office I, Korea Carbon Industry Promotion Agency, Jeonju 54852, Korea; rkdtjdgus95@gmail.com (S.-H.K.); kkw01090063344@gmail.com (K.-W.K.); 2Department of Carbon Materials & Fiber Engineering, Jeonbuk National University, Jeonju 54896, Korea; 3Department of Carbon and Nano Materials Engineering, Jeonju University, Jeonju 55069, Korea

**Keywords:** high-density polyethylene (HDPE), carbon fiber, cross-linking, electron beam (E-beam), sulfonation

## Abstract

In this study, a method for manufacturing high-density polyethylene (HDPE)-based carbon fibers using a hybrid cross-linking method was studied. HDPE precursor fibers were first cross-linked with an electron beam (E-beam) at an irradiation dose of 1000–2500 kGy, and then cross-linked in sulfuric acid at 80–110 °C for 60 min. Hybrid crosslinked fibers were carbonized for 5 min at a temperature of 900 °C. As a result, the hybrid crosslinked fibers had a carbonization yield of 40%. In addition, the carbonized fibers after hybrid crosslinking exhibited perfect fiber morphology, and HDPE-based carbon fibers with (002) and (10*l*) peaks, which are the intrinsic XRD peaks of carbon fibers, were successfully prepared.

## 1. Introduction

Carbon fiber is a lightweight, high-tech material composed of 92% carbon atoms with a fine graphite crystal structure and excellent mechanical, thermal, and electrical properties. It has various military, aviation and space, automobile, ships, and sports or leisure applications. In addition, carbon fiber can either be pitch-based, rayon-based, or polyacrylonitrile (PAN)-based [1,2,3,4]. Pitch-based carbon fiber is cheaper than PAN-based carbon fiber but offers lower stiffness than the latter [5]. Rayon-based carbon fiber is manufactured using rayon fiber as a precursor. However, it is not widely commercialized because of the difficulty in controlling the precursor’s physical properties and its low carbonization yield [6]. PAN is considered the optimal precursor when producing carbon fiber because it has better mechanical properties than pitch or rayon and has a potentially extensive commercial use. However, producing PAN-based carbon fiber incurs high production costs, making cost reduction one of the challenges to overcome [7,8,9]. Figure 1 shows the pricing structure of PAN-based carbon fiber. Precursor accounts for the largest percentage of the total manufacturing cost, i.e., 51%. Therefore, the most effective option for lowering production cost is to replace the precursor that takes up the most significant manufacturing cost-share [10,11]; possible candidates for precursor materials include lignin [12], polypropylene (PP) [13], and polyethylene (PE) [9,11,14]. Additionally, composite fibers such as PAN/nanocellulose and PAN/lignin that utilize the superior mechanical properties of PAN and use renewable resources (e.g., lignin and cellulose) have been studied [15,16]. Among the following options, PE can form long fibers through the melt spinning process. PE fiber costs EUR 1–1.5/kg, lower than PAN fiber (EUR 2–5/kg) [9]. However, being a thermoplastic material, PE has low thermal properties, but its thermal properties can be enhanced by cross-linking its chain structure, stabilizing it into a ring structure. Cross-linking stabilization methods include silane cross-linking, peroxide cross-linking, electron-beam cross-linking, and sulfuric acid cross-linking [17,18,19,20]. Silane cross-linking offers the benefit of production without substantial investment in facilities. However, due to its weak physical properties, it does not significantly affect thermal stability [21]. Peroxide cross-linking uses liquid or solid peroxide as a chemical cross-linking agent, allowing materials to cross-link in a molten state at high temperatures [22]. In comparison, electron-beam cross-linking is difficult to use on its own, as it offers a low level of cross-linking density. Sulfuric acid cross-linking has been confirmed to manufacture stable carbon fiber through cross-linking at 180 °C or higher. However, it poses critically hazardous environmental conditions and inflicts thermal shock to PE, altering the fiber’s shape [20]. Based on these disadvantages, a method that allows for sulfuric acid cross-linking at temperatures that do not exceed 100 °C is required. Thus, this study proposes improving the PE-based carbon fiber manufacture using a hybrid cross-linking method that combines electron beam pretreatment and sulfuric acid cross-linking.

## 2. Experiment Details

### 2.1. Materials

This study utilized high-density polyethylene (HDPE) as a precursor material to manufacture carbon fiber (HDPE, 2700J, LG chem, Seoul, Korea), and the material was subject to melt spinning (Melting index = 7.0 g/10 min) to create a fiber with a 15–20 μm diameter. Sulfuric acid (98% H_2_SO_4_, Daejung chem, Siheung-si, Korea) was used as a cross-linking agent.

### 2.2. Manufacturing the Stabilized Fiber Using Hybrid Cross-Linking

The HDPE fiber was cut into 30 cm-long pieces, and hybrid cross-linking was performed using electron beam irradiation and sulfuric acid cross-linking. The fiber was irradiated with an electron beam (2.5 MeV, 10 m/min) at 1000 kGy, 1500 kGy, 2000 kGy, and 2500 kGy, using an electron beam irradiator for the primary cross-linking. For the secondary cross-linking (hybrid cross-linking), after the primary cross-linking, the HDPE fiber was dipped in sulfuric acid and heated to 80 °C, 90 °C, 100 °C, and 110 °C while maintaining a certain degree of tension, and cross-linked for 60 min at the temperatures mentioned above. The hybrid cross-linked HDPE fiber was cleaned with distilled water and sufficiently dried for 24 h in a dry oven (60 °C). Table 1 lists the HDPE samples with varying cross-linking conditions.

### 2.3. Carbon Fiber Manufacturing

The hybrid cross-linked HDPE fiber was carbonized to manufacture carbon fiber. A thermogravimetric furnace was used to raise the temperature from room temperature (25 °C) to 900 °C at 2 °C/min in a nitrogen gas atmosphere (300 mL/min). The temperature was maintained for 5 min and passively cooled.

### 2.4. Thermal Properties of the Stabilized Fiber

Differential scanning calorimetry (DSC-60, SHIMADZU, Kyoto, Japan) and thermogravimetric analysis (TGA-50, SHIMADZU, Kyoto, Japan) were utilized to observe changes in the HDPE fiber’s thermal properties before and after stabilization, and thermal properties were analyzed in a nitrogen atmosphere (50 mL/min). In addition, two DSC analysis cycles were performed in a 30–300 °C temperature range and at a heating and cooling rate of 20 °C/min, while the TGA analysis was performed in a 30–900 °C temperature range and at a heating rate of 20 °C/min.

### 2.5. Morphology of the Stabilized Fiber and the HDPE-Based Carbon Fiber

The manufactured carbon fiber’s surface shape was observed with a scanning electron microscope (SEM, AIS2000C, Seron Tech., Uiwang-si, Korea). In addition, the manufactured carbon fiber’s surface was observed after coating it with platinum (Pt) to minimize the charging effect caused during SEM analysis, and the analyzer chamber’s basic pressure was approximately 5.0 × 10^−5^ torr, while the acceleration voltage was 20 kV. Changes in the HDPE fiber’s fine structure and the manufactured carbon fiber before and after stabilization were analyzed using X-ray diffraction (XRD, MiniFlex 600, Rigaku Co., Tokyo, Japan) in a 10–90° range at 4°/min, using Cu-Kα radiation.

### 2.6. Tensile Properties

Before and after stabilization, the HDPE fiber’s tensile properties were measured using a universal testing machine (UTM, ST-1000, Salt Co., Incheon, Korea) based on a single fiber tensile test (ASTM C 1239-07). The fiber’s gauge length was 25 mm, and the crosshead speed was 0.3 mm/min. Twenty pieces were measured per sample to calculate the mean value.

## 3. Findings and Discussion

### 3.1. HDPE Fiber after Primary Cross-Linking

#### 3.1.1. Calorimetry

Figure 2 and Table 2 show the results of DSC analysis of HDPE fibers after primary crosslinking using electron beam irradiation. As shown in Figure 2, after the primary cross-linking, the HDPE fiber’s heat absorption and caloric value decreased from −26.59 and 26.76 cal/g to −18.75 and 17.5 cal/g, respectively. This result confirmed that electron beam irradiation contributed to the cross-linking of the HDPE fiber. In addition, as shown in Table 2, the temperature spot of T_m_ and T_c_ decreased along with electron beam irradiation, from 131.45 °C and 101.6 °C to 122.51 °C and 88.67 °C, respectively. These results suggest that, due to the increase in electron beam irradiation, the not fully cross-linked molecular chains existed in thermally unstable states, resulting in melting and crystallization at a lower temperature.

#### 3.1.2. X-ray Diffraction Analysis

Changes in the HDPE fiber’s structure after primary cross-linking with electron beam irradiation were analyzed using XRD. Table 3 presents a detailed view of the changes using the Scheerer equation. Here, θ is the Bragg angle, FWHM is the full width at half maximum, d_110_ and d_200_ are planar spacings of (110), (200), and L_a_ is the crystal size (length) [23]. As represented in Table 3, L_a_ declined from 11.1 and 23.3 nm to 9.6 and 19.5 nm at the electron beam irradiation of 100 kGy, and L_a_ increased as irradiation increased to 2500 kGy. The findings could be attributed to electron beam irradiation on the severance and cross-linking of the C–C and C–H bonding of the HDPE fiber’s molecular chains [24]. Moreover, L_a_ declined as C–C bonds broke up to 1000 kGy. However, the severed bonds were restored at higher irradiation levels, increasing L_a_.

#### 3.1.3. Tensile Properties

Figure 3 shows changes in the HDPE fiber’s tensile properties after primary cross-linking with electron beam irradiation. When electron beam irradiation increased on the HDPE fiber, the tensile strength declined from 1.09 GPa to 0.13 GPa, and the tensile modulus increased from 8.45 GPa to 19.09 GPa. The decline in tensile strength was likely caused by the insufficiently cross-linked or severed unstable molecular chains, while the increase in tensile modulus could be attributed to the molecular structures’ cyclization by cross-linking. Moreover, the findings related to the tensile properties suggest that an HDPE fiber irradiated with an electron beam at 1000 kGy was likely to maintain its shape even after secondary cross-linking and carbonization, and an HDPE fiber irradiated below 1000 kGy was unsuitable for carbonization because of its cross-linking density. Therefore, secondary cross-linking was performed using HDPE fibers cross-linked at 1000–2500 kGy.

### 3.2. Hybrid Cross-Linked HDPE Fiber

#### 3.2.1. Thermal Properties

DSC and TGA were used to verify the hybrid cross-linking HDPE fiber’s thermal properties under various conditions (Figure 4 and Figure 5). In the DSC analysis represented in Figure 4, all hybrid cross-linked HDPE fibers showed a greater calorie decline than electron beam irradiation (Figure 2a). This observation suggests that electron beam irradiation increased radical formation in the polymer chains, accelerating cyclization during sulfuric acid cross-linking. In addition, calories declined as sulfuric acid treatment temperatures increased. No change in calories was observed in the HDPE fiber hybrid cross-linked at 100 °C or higher, indicating that all HDPE fibers were cyclized by the hybrid cross-linking and became insoluble. The TGA analysis findings in Figure 5 show a steady decline in disintegration at the PE disintegration temperature range (440–520 °C), as electron beam irradiation and sulfuric acid temperature increased. Moreover, carbonization yield increased. As seen from the DSC findings, increasing electron beam irradiation and sulfuric acid temperature accelerated cyclization lowered PE disintegration and pushed the carbonization yield. Mass declined faster at lower temperatures (<450 °C), suggesting that lower temperatures increased partially cross-linked unstable molecules. The fiber hybrid cross-linked at 110 °C or higher was expected to maintain its shape even after carbonization based on findings related to the thermal properties. Therefore, additional experiments were conducted by varying the sulfuric acid treatment time from 30–120 min, with 30 min intervals, and the TGA results are shown in Figure 6. As shown in Figure 6, a mass reduction was observed at the PE disintegration temperatures when the fiber was treated for 60 min or less. However, no mass reduction was observed in fibers treated for 90 min or more, which may be attributed to the fully cross-linked PE fiber.

#### 3.2.2. X-ray Diffraction Analysis

X-ray diffraction patterns were analyzed to verify changes in the HDPE fiber’s crystalline properties cross-linked in sulfuric acid at 110 °C for 30 min to 120 min (Figure 7). With an increase in sulfuric acid treatment time, the materials’ (110) and (200) peak strength declined. However, no (110) and (002) peak was observed when the fiber was treated for 90 min or longer. This observation can be attributed to an increase in sulfuric acid treatment time, which raised the cross-linked density, destroyed HDPE’s crystalline areas, and increased the non-crystalline areas.

### 3.3. Carbonized Fiber

Surface Properties

Figure 8 shows the post-carbonization cross-sectional SEM photograph of the samples subject to 1000 kGy electron beam irradiation and sulfuric acid treatment at 110 °C. Figure 8a,b show that, as sulfuric acid treatment time increased, internal voids decreased until it was no longer observed in Figure 8c, indicating that cross-linking progressed externally to internally. In addition, hollow fiber shapes are seen in Figure 8a,b, meaning the inside was disintegrated during carbonization because only the outer areas were cross-linked. In contrast, perfect fiber shapes can be seen in Figure 8c,d because the materials were sufficiently cross-linked to the internal area. Lastly, Figure 9 shows the XRD graph of the carbonized HDPE fiber after hybrid cross-linking, showing the (002) and (10*l*) peaks in carbon fiber.

## 4. Conclusions

In conclusion, this study successfully manufactured carbon fiber using HDPE as a precursor. Hybrid cross-linking, which combined electron beam irradiation and sulfuric acid cross-linking, was suitable for creating high carbonization yield and maintaining fiber shape while raising the cross-linking density. In the DSC analysis, increasing the irradiation dose of the electron beam pretreatment and the temperature increase of the sulfuric acid crosslinking accelerated the stabilization of HDPE. The sample yields were affected by all conditions, including irradiation, sulfuric acid treatment temperature, and duration based on the TGA analysis. In addition, in the XRD analysis, electron beam treatment and sulfuric acid cross-linking affected HDPE’s crystalline behavior. After carbonization, (002) and (10*l*) peaks were observed in the carbon fiber. The findings demonstrate that the conditions used in this study lowered the sulfuric acid cross-linking temperature and affirmed the potential of HDPE as a carbon fiber precursor. In the future, this research team plans to manufacture HDPE-based activated carbon fiber by applying hybrid cross-linked HDPE-based carbon fiber, which has confirmed its potential as a carbon fiber precursor.

## Figures and Tables

**Figure 1 polymers-13-02157-f001:**
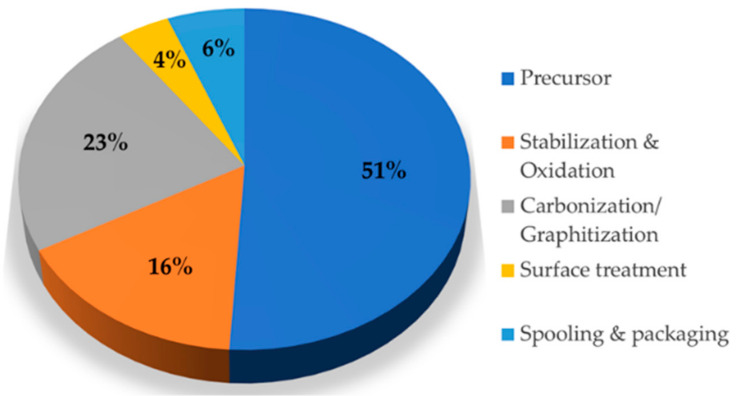
Polyacrylonitrile-based (PAN-based) carbon fiber price structure [10].

**Figure 2 polymers-13-02157-f002:**
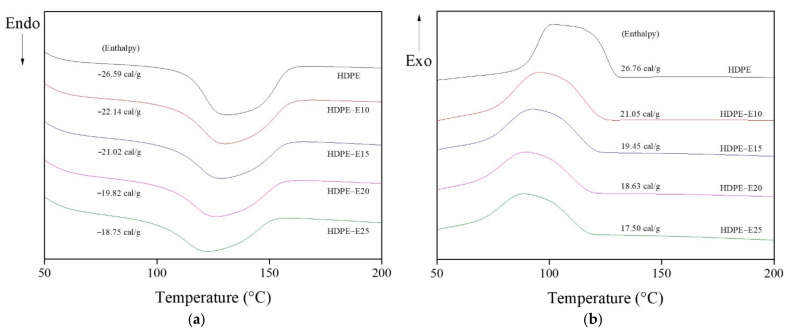
Differential scanning calorimetry (DSC) measurement result of a cross-linked high-density polyethylene (HDPE) fiber after electron beam irradiation: (**a**) heating and (**b**) cooling scan.

**Figure 3 polymers-13-02157-f003:**
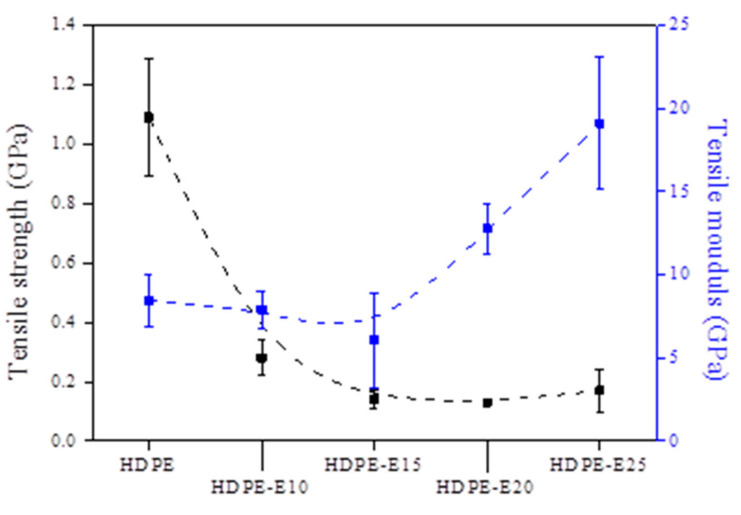
Tensile strength and tensile modulus results of cross-linked high-density polyethylene (HDPE) fibers after electron beam irradiation.

**Figure 4 polymers-13-02157-f004:**
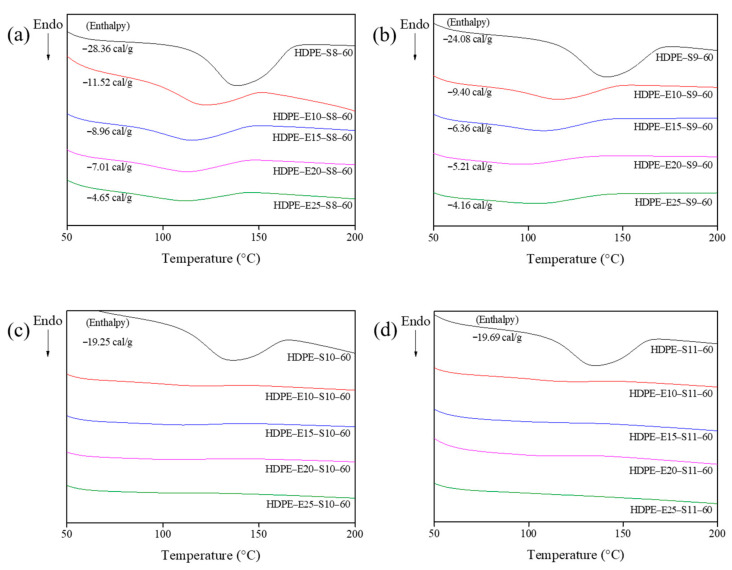
Differential scanning calorimetry (DSC) measurement results of high-density polyethylene (HDPE) fibers secondary cross-linked for 60 min at various sulfuric acid temperatures after electron beam cross-linking: (**a**) 80 °C, (**b**) 90 °C, (**c**) 100 °C, and (**d**) 110 °C.

**Figure 5 polymers-13-02157-f005:**
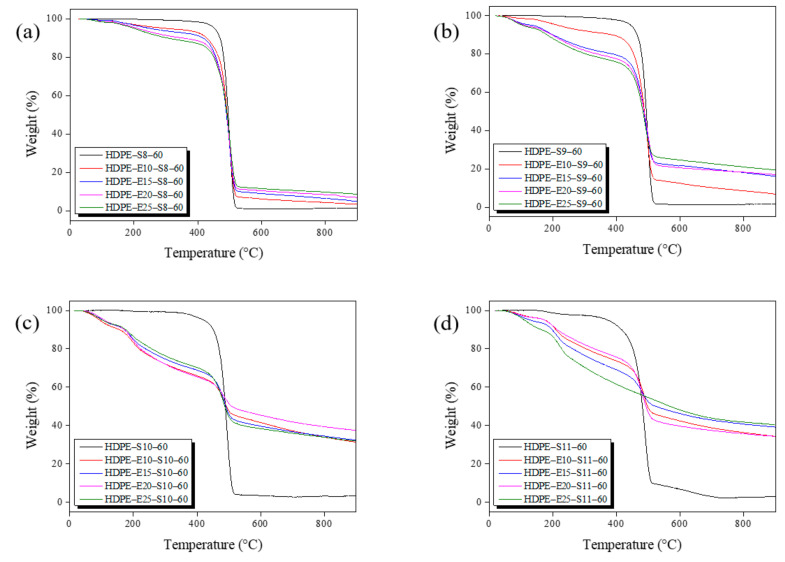
Thermogravimetric analysis (TGA) measurement results of high-density polyethylene (HDPE) fibers secondary cross-linked for 60 min at various sulfuric acid temperatures after electron beam cross-linking: (**a**) 80 °C, (**b**) 90 °C, (**c**) 100 °C, and (**d**) 110 °C.

**Figure 6 polymers-13-02157-f006:**
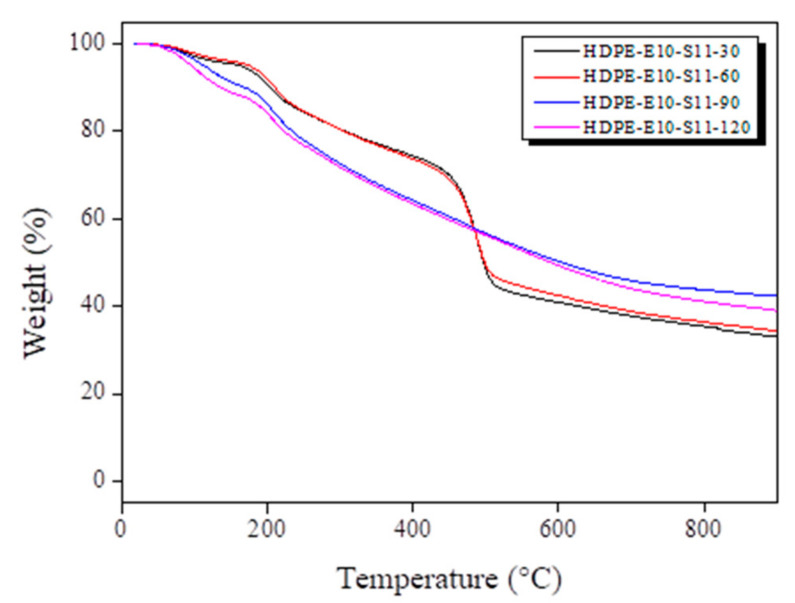
Hybrid cross-linked high-density polyethylene (HDPE) fiber thermogravimetric analysis (TGA) curves at various times.

**Figure 7 polymers-13-02157-f007:**
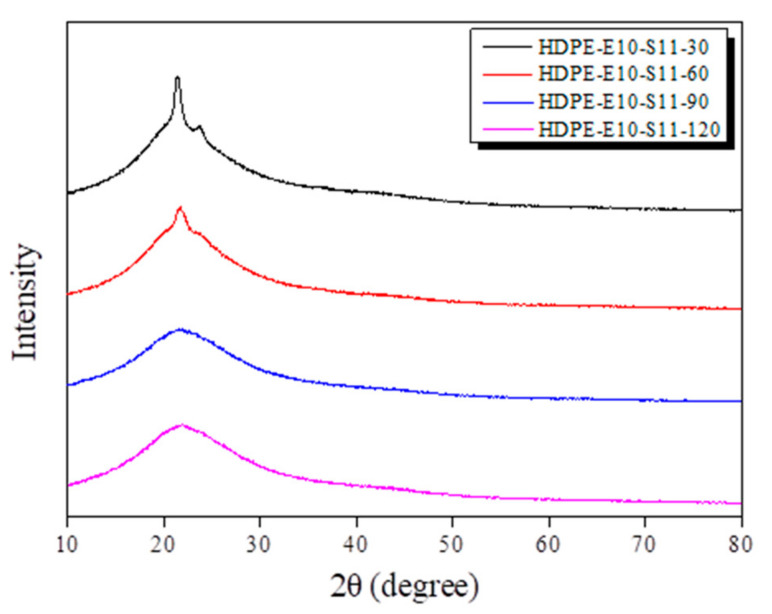
Hybrid cross-linked high-density polyethylene (HDPE) X-ray diffraction (XRD) spectra at various times and at 110 °C.

**Figure 8 polymers-13-02157-f008:**
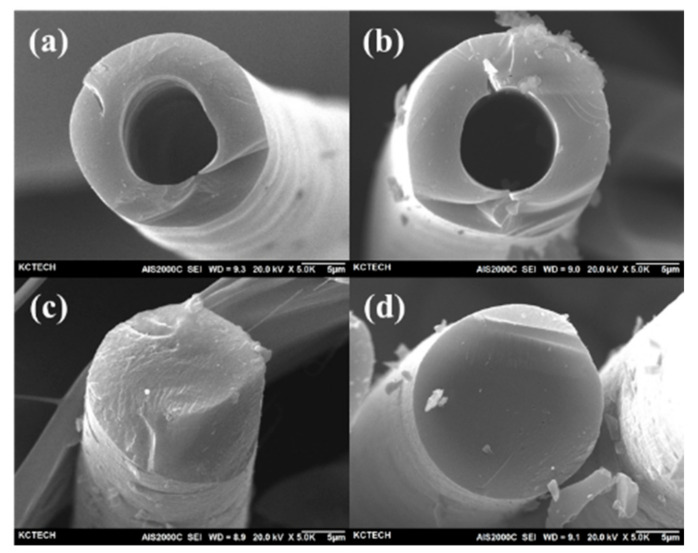
Scanning electron microscope (SEM) cross-section after carbonization of cross-linked high-density polyethylene (HDPE) fibers at 110 °C and at various cross-linking times: (**a**) 30 min, (**b**) 60 min, (**c**) 90 min, and (**d**) 120 min.

**Figure 9 polymers-13-02157-f009:**
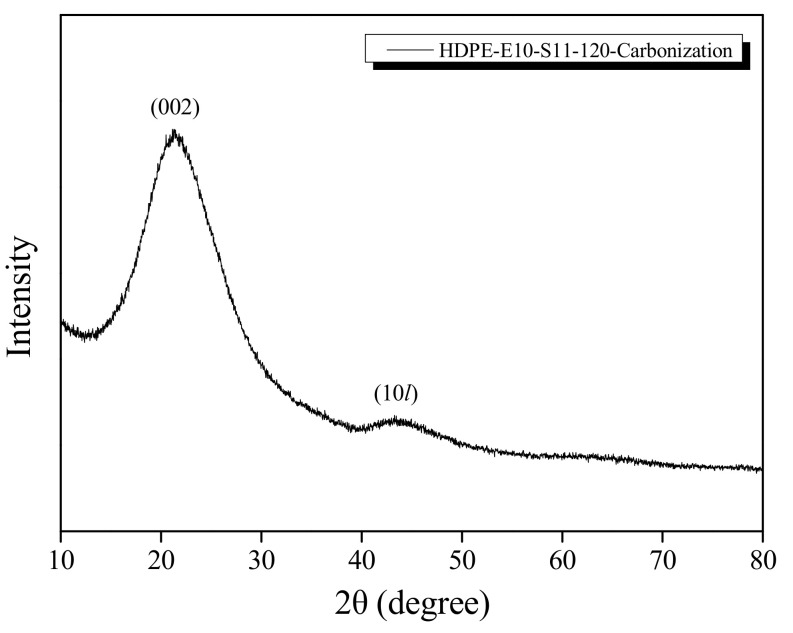
High-density polyethylene-based (HDPE-based) carbon fiber X-ray diffraction (XRD) spectra.

**Table 1 polymers-13-02157-t001:** Cross-linking condition and sample naming.

	Cross-Linking Conditions		
Sample	Electron Beam	Sulfuric Acid	
	Irradiation (kGy)	Temp (°C)	Time (min)
HDPE	As-received		
HDPE-E10 ^a^	1000	-	-
HDPE-E10-S8-60 ^b^	1000	80	60

^a^ E10 means that the electron beam irradiation was carried out at 1000 kGy. ^b^ S8-60 means that sulfuric acid cross-linking was carried out in sulfuric acid at 80 °C for 60 min.

**Table 2 polymers-13-02157-t002:** Differential scanning calorimetry (DSC) measurement result of electron beamed cross-linked high-density polyethylene (HDPE) fiber.

Sample	Onset T_m2_ (°C)	Peak T_m2_ (°C)	Onset T_c2_ (°C)	Peak T_c2_ (°C)
HDPE	114.74	131.45	129.18	101.60
HDPE-E10	113.08	132.27	121.16	94.71
HDPE-E15	109.74	128.54	119.43	92.45
HDPE-E20	107.63	126.25	116.61	89.64
HDPE-E25	103.48	122.51	116.18	88.67

**Table 3 polymers-13-02157-t003:** X-ray diffraction (XRD) results of cross-linked high-density polyethylene (HDPE) fibers after electron beam irradiation.

Sample	110 Peak				200 Peak			
	2θ	FWHM (2θ)	d_110_ (Å)	L_a_ (Å)	2θ	FWHM (2θ)	d_200_ (Å)	L_a_ (Å)
HDPE	21.41	0.73	4.15	111.11	23.58	0.71	3.77	232.98
HDPE-E10	21.38	0.84	4.15	96.35	23.70	0.85	3.75	194.96
HDPE-E15	21.42	0.77	4.14	105.47	23.72	0.80	3.75	207.18
HDPE-E20	21.26	0.67	4.18	119.82	23.54	0.69	3.78	239.30
HDPE-E25	21.32	0.68	4.16	118.82	23.58	0.71	3.77	232.98

## Data Availability

The data presented in this study are available on request from the corresponding author.

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
