# Peer review of "Carbon Fibers from High-Density Polyethylene Using a Hybrid Cross-Linking Technique"

_polymers, 2021, doi:10.3390/polym13132157_

Round 1
Reviewer 1 Report
It is interesting to deveop new metods to reduce the cost of carbonfiber. The authors tried to evaluat the effectiveness of crosslinking, however, the problem to this article is that the authors did not accurately explain thsese data.
The authors introduced the method of preparing and analysing the carbon fiber, however, the readers would also want to know the results. The performance of the carbon fiber should be given.
The title of 2.6 should be “mechanical property” or “tensile property”, not physical property.
How to measure carbon content?
The meaning of Figure 1 is not clear.
What’s the potential effect of cryastal size change?
“The hybrid–cross-linked HDPE fiber was carbonized to manufacture carbon fiber.” My question is that as control, whether the untreated HDPE was carbonized? It should be explained in Experiment section.
All the figures and tables give the “measurement result of electron beamed cross- linked high-density polyethylene (HDPE) fiber”, but how about carbon fiber? Did authors test carbon fibers?
Based on the data in Figure2, Figure3 and Table 2, the tensile strength and thermal stability of the cross-linked fiber decreased comparing to HDPE, so what’s the advantage of present study?
How do the properties of the carbon fibers in this study comparing with conventional carbon fibers?
Author Response
Thank you for your kind comments. Added answer based on your comments. Please refer to the attached file.

Reviewer 2 Report
The manuscript of Seong-Hyun Kang et al "Carbon Fibers from High-Density Polyethylene using a Hybrid Cross-Linking Technique" is dedicated to the production of carbon fibers from HDPE precursors.
At the moment, the work is not yet ready for publication. Authors should reflect in the literature review the results already obtained by the scientific community on this issue. In the Introduction, in my opinion, it makes sense to mention a new type of composite CF precursors, namely, based on cellulose with PAN. Clearly formulate a goal and highlight the novelty of the research.
I recommend that the authors change the abstract, remove the enumeration of the methods used and add concrete results.
Line 109. it is better to change this expression "Figure 2 and Table 2 show the DSC calorie analysis"
Line 186. It is desirable to change the colors on the diagram.
The advantages of the new precursors, which were mentioned at the beginning of the manuscript, are gradually fading into the background. This requires non-trivial ways to prepare precursors. The carbon residue values do not exceed the values for PAN precursors and, in some cases, for cellulose fibers. The resulting structure for CF will most likely not provide good mechanical properties. And in my opinion, it should be added to the manuscript.
Author Response
Thank you for your kind comments. According to your comment, we added answer.
Please see the attachment.
